# Impact of Reference Data Sampling Density for Estimating Plot-Level Average Shrub Heights Using Terrestrial Laser Scanning Data

**Aaron E. Maxwell** [1,*] , **Michael R. Gallagher** [2] , **Natale Minicuci** [3] , **Michelle S. Bester** [1] , **E. Louise Loudermilk** [4] , **Scott M. Pokswinski** [5] **and Nicholas S. Skowronski** [6]

1 Department of Geology and Geography, West Virginia University, Morgantown, WV 26506, USA
2 USDA Forest Service, Northern Research Station, New Lisbon, NJ 08064, USA
3 Tall Timbers Research Station, New Lisbon, NJ 08064, USA
4 USDA Forest Service, Southern Research Station, Athens, GA 30602, USA
5 USDA Forest Service, Northern Research Station, Morgantown, WV 26506, USA
6 New Mexico Consortium, Los Alamos, NM 87544, USA
* Correspondence: aaron.maxwell@mail.wvu.edu

**Abstract:** Terrestrial laser scanning (TLS) data can offer a means to estimate subcanopy fuel characteristics to support site characterization, quantification of treatment or fire effects, and inform fire modeling. Using field and TLS data within the New Jersey Pinelands National Reserve (PNR), this study explores the impact of forest phenology and density of shrub height (i.e., shrub fuel bed depth) measurements on estimating average shrub heights at the plot-level using multiple linear regression and metrics derived from ground-classified and normalized point clouds. The results highlight the importance of shrub height sampling density when these data are used to train empirical models and characterize plot-level characteristics. We document larger prediction intervals (PIs), higher root mean square error (RMSE), and lower R-squared with reduction in the number of randomly selected field reference samples available within each plot. At least 10 random shrub heights collected in situ were needed to produce accurate and precise predictions, while 20 samples were ideal. Additionally, metrics derived from leaf-on TLS data generally provided more accurate and precise predictions than those calculated from leaf-off data within the study plots and landscape. This study highlights the importance of reference data sampling density and design and data characteristics when data will be used to train empirical models for extrapolation to new sites or plots.

**Keywords:** forest fires; prescribed forest fires; terrestrial laser scanning; TLS; fire effects; fire fuels; fuel load; forest understory characterization

## 1. Introduction

Forest fires have large impacts on ecosystems, economies, infrastructure, and society [1,2]. Climate change and other anthropogenic landscape alterations have and will continue to impact the spatial variability, intensity, and patterns of fire occurrence and associated effects; further, urbanization and development within fire-prone areas will put more infrastructure, property, and individuals at risk [1–3]. Determining appropriate treatments and prescribed fire practices to meet land management objectives [4–7] and predicting fire behavior and associated primary and secondary effects using modeling techniques [8–10] require estimation of site characteristics including wind conditions [11,12], moisture content [13,14], and fuel availability [15–17]. Accurate estimation of such characteristics is necessary to support decision making related to when and how to administer management (i.e., thinning or prescribed fires), to assess the results of prescribed fire and other fuel treatment options in the context of management goals, and to predict how a fire is likely to progress across the landscape. This can aid in managing the fire, improving the safety of field crews,

and containing the event. Quantifying fuels is particularly important since they can be manipulated and managed via treatment practices, including prescribed fire and stand thinning, in contrast with moisture content and weather conditions [15,18,19].

In the context of prescribed fire and land management, practitioners often need to quantify the reduction in the amount of fuels following a prescribed fire or treatment activity, which requires a means to measure or estimate fuel material characteristics consistently and repeatedly [18]. Physical process-based fire behavior models must characterize fuels using a finite set of variables [1]; for example, models that make use of the equations of Rothermel [20] characterize fuels based on loading (i.e., weight of fuel per unit landscape area), particle size distributions, and fuel bed depths for multiple strata. However, there are many characteristics of individual fuel particles (e.g., thermal conductivity, particle density, thermal diffusion, ash content, moisture content, size, thickness, and shape) and aggregated fuel beds (e.g., particle size distribution, bulk density (weight per volume of biomass), packing ratio, porosity, depth, vertical stratification, patch size, and live vs. dead fractions) that can impact fire behavior. Thus, detailed, accurate, and spatially explicit characterization of fuels is limited, despite the importance of such information in management and predictive modeling [9,17,21].

Traditional field-observer methods of collecting basic fuels and fire effects data at the plot-level have been increasingly foregone for remote sensing approaches, such as multi-spectral scanning (MSS) and airborne light detection and ranging (LiDAR) systems (ALS), that gather wall-to-wall information with unmatched speed and repeatability [22,23]. These remote sensing approaches excel at predicting bulk fuel loading, coarse-scale fire effects on vegetation and substrates, and detailed changes in forest canopy structure [24]; yet despite these advantages, there are important fundamental disadvantages compared with field methods that have not yet been resolved and which drive a continued interest. First, due to occlusion by canopy vegetation, MSS and ALS predictions are often weak at predicting understory vegetation conditions that frequently define the objectives and fire behavior expectations of prescribed fires in forest and woodland settings [25,26]. Simulations via processed-based fire behavior modeling have illustrated that these understory fuels that are modified by prescribed burning can play a significant role in driving spread rates during large wildfires [27]. Lastly, although some MSS and ALS sensors can be tasked for timing specific to management needs, expense and logistical challenges drive opportunistic use of data collected for other purposes to dominate instead of planned data collections that are timed to provide the data that will best match the timing of fire occurrence and associated plant phenology for fuels and fire effects predictions [28,29].

One way to potentially increase the efficiency and repeatability of plot-scale fuel material characterization compared with traditional manual measurement methods is to use terrestrial laser scanning (TLS) [30,31]. Such instruments use active remote sensing via laser pulses to generate a dense sample of point measurements to characterize a three-dimensional space [30,31]. In this study, we specifically explore the prediction of plot-level mean shrub heights (i.e., shrub fuel bed depth) using a set of single-date, single-location, single-return TLS point clouds collected using affordable units within the Pinelands National Reserve (PNR) region of the state of New Jersey, United States (USA). Our primary goal is to assess how well shrub heights are empirically predicted using a set of TLS-derived metrics as predictor variables when (1) varying the number of randomly selected shrub height measurements within the plot used to approximate the average height to train models and (2) using metrics derived from leaf-off vs. leaf-on TLS data.

There is a notable lack of research associated with assessing affordable units and comparing results to traditional field methods. Stovall and Atkins [32] offered a comparison of two affordable units, but did not compare the results to field reference data. Pokswinski et al. [33] outlined a field methodology incorporating affordable TLS data, but also did not provide a comparison to field reference data. Thus, we argue that this study is of value as it offers comparisons of a key subcanopy fuel load measure, which has been traditionally estimated with transect- or point-based field methods, with single-scan, single-return TLS

data, which can be easily and quickly collected to replace or augment existing methods. In other words, we assess TLS data in the context of operational adoption.

## 2. Background

It is not possible to directly measure the large set of fuel particle and fuel bed characteristics that may impact fire behavior, especially when such data are needed over large spatial extents and the fuel characteristics are spatially heterogeneous. As a result, field sampling methods have been developed to collect key information needed to assess the efficacy of treatments, as input to fire behavior models, and to generally inform decision making [34,35]. Further, these field measurements are often combined with statistical methods and species-specific allometric equations to estimate measures of interest, such as bulk density or fuel load [1,36].

All field methods have strengths and weaknesses. Sikkink et al. [37] compared the fixed-area plot, planar transect, photoload, photoload microplot, and photo series methods for estimating loadings for different fuel components. The planar transect method was generally suggested to be the best method based on multiple criteria. Keane et al. [38] compared planar intercept, fixed-area microplot, and photoload methods. They noted the fixed-area microplot method's accuracy and the need for intensive sampling to accurately estimate fuel biomass regardless of the chosen method. There can also be issues of repeatability even when the same field methods are used over time; for example, Westfall and Woodall [39] assessed the repeatability of large-scale forest fuel sampling conducted as part of the Forest Inventory and Analysis (FIA) program of the United States Department of Agriculture (USDA) Forest Service and documented failure to obtain desired levels of repeatability in more than half of the measured attributes. Further, one-third of the attributes exhibited measurement bias; however, bias was less problematic when results were aggregated to the plot-level [39]. It is important to note that field methods are not standardized or consistently collected, which can limit comparisons across field campaigns and studies [1,30,40]. Further, different agencies and regions have developed disparate protocols to meet their specific needs. For example, the Department of Sustainability and Environment within the Victoria, Australia (AU) government, has developed protocols (see Hines et al. [35]) that are different from those implemented by the USDA Forest Service (see Prichard et al. [41]). In summary, collecting accurate, consistent, and spatially explicit fuel measurements at desired spatial resolutions is a complex and time-consuming task, which is of concern since fuel characterization is central to management and modeling.

Given the issues associated with field-based methods for characterizing fuels, the need for spatially explicit, voxelized representations of fuel characteristics consistently across large spatial extents, and the necessity to capture spatial variability, remote sensing data and methods have been explored for estimating fuel characteristics. Remote sensing is attractive for such mapping problems as it allows for estimating parameters at the scale of individual pixels, or extended into 3D space using voxels, and potentially across large spatial extents [22]. As an added benefit, collecting and updating remotely sensed data is generally less costly and time intensive than undertaking multiple field campaigns [42]. For a recent review of remote sensing techniques for characterizing fuels, see Gale et al. [22]. Given that hyperspectral and multispectral data offer limited information regarding the vertical structure of the forest canopy, light detection and ranging (LiDAR) has been integrated with spectral data (e.g., [43–45]) or used independently (e.g., [46,47]) to assess and characterize canopy fuels.

Aerial LiDAR allows for the collection of multiple returns from a single laser pulse, resulting in some degree of canopy penetration and subcanopy characterization [48]. For example, Skowronski et al. [49] proposed a consistent method for characterizing hazardous fuels at the wildland–urban interface in New Jersey, USA based on the integration of aerial LiDAR, aerial imagery, and cadastral datasets. Erdody and Moskal [45] integrated LiDAR and high spatial resolution aerial near-infrared (NIR) imagery for estimating canopy height, base height, bulk density, and available fuels in Washington, USA and documented the

value of LiDAR. They suggested only slight improvements when combining the imagery data and LiDAR in comparison with just using the LiDAR data. Other studies have relied on only LiDAR; for example, Skowronski et al. [50] assessed the characterization of forest biomass using LiDAR and documented varying correlations between LiDAR-obtained heights and biomass within forest community types; they found that binning data into height bins was useful for estimating the presence of ladder fuels [50].

Although LiDAR can aid in characterizing the vertical structure of the tree canopy, due to a limited number of returns reaching the lower strata and the confounding impacts of an often heterogeneous canopy, subcanopy vegetation, shrubs, and downed woody debris are generally not well characterized [17,46,51,52]. Subcanopy fuels occurring in the shrub, litter, and duff layers and downed woody debris often constitute a large fraction of fuel materials within forest stands and play a significant role in determining fire behavior [9,21,52]; thus, the inability to characterize subcanopy fuel loads and spatial patterns is a key knowledge gap in management and fire behavior modeling. Even for canopy fuels, Skowronski et al. [46] documented the value of combining downward scanning aerial LiDAR and upward sensing profiling LiDAR to obtain a better characterization of the three-dimensional canopy structure in comparison with only using aerial data. Many studies note the need for better estimating subcanopy fuel loads and spatial patterns and developing more accurate three-dimensional fuel models (e.g., [9,10,53,54]). Arroyo et al. [42] noted the value of combining data from multiple remote sensing sources to better characterize fuels.

TLS data offer the ability to collect data beneath the forest canopy at a high spatial resolution and, thus, can complement aerial- or satellite-based remote sensing for site-level characterization [51,52,55,56]. Table 1 below summarizes studies that have assessed TLS for measuring fuels, or biomass more generally. As the table highlights, TLS has been used to estimate a wide variety of fuel-related parameters, both in the lower strata and in the tree canopy, and has been compared with a wide range of field methods. Field methods that have been used for comparison include both destructive methods, such as clip plots (e.g., Rowell et al. [53,54,57]), and non-destructive point- or planar-intercept methods (e.g., Loudermilk et al. [52] and Alonso-Rego et al. [51]). Many studies have focused on the analysis of TLS data collected using analytical-grade units with sites characterized using scans from multiple locations to more densely sample the site and minimize the impact of occlusion (e.g., Garcia et al. [56], Loudermilk et al. [17], and Rowell et al. [53,54,57]). Few studies have compared pre- and post-event TLS data for assessing biomass loss, fuel treatment effects, or burn severity. Notable exceptions are Hudak et al. [58], who estimated changes in occupied voxel density and shrub fuel bulk density, and Gallagher et al. [59], who compared changes in TLS metrics to the composite burn index (CBI) [60], a field-based measure of burn severity based on visual site assessment.

Many modeling techniques used to make estimates of landscape characteristics from remotely sensed data, such as linear regression and many machine learning algorithms, rely on supervised learning or empirical methods, highlighting the importance of reference data availability and quality [48]. Reference data quality has been noted to be of particular importance when estimating forest attributes, such as aboveground biomass, with the impact of sample size varying between methods and algorithms [61]. Reference data collection techniques are of specific concern when the goal is the collection of standardized inventories and mapping over large spatial extents, such as national-level forest inventories [62], since sampling methods and sample placement can impact resulting models [63]. Other than training models, reference data are also required to assess model performance, and biased or uncertain validation samples can hinder meaningful assessment of model outputs [64,65]. Thus, there is a need for research that assesses the impact of reference data density on resulting model performance across spatial scales and for varying mapping or modeling tasks.

**Table 1.** Summary of studies that have compared TLS data to field-based methods for characterizing fuels.

| Study | TLS Data | Ground Data | Parameters | Landscape |
|---|---|---|---|---|
| Loudermilk et al. (2009) [52] | Multiple scans; ILRIS | Point-intercept; fuel bed and litter depth; presence/absence of fuel; vegetation type (non-destructive) | Leaf biomass; leaf area; point-intercept volume | Longleaf pine (Georgia, USA) |
| Garcia et al. (2011) [56] | Multiple scans from the same position but rotated; Riegl LMS-Z390i | DBH; crown diameter, height, and base height; planar transects (non-destructive) | Canopy height, cover, and base height; fuel strata gap | Scots pine, larch, and mixed oak/birch (Cheshire, UK) |
| Loudermilk et al. (2012) [17] | Multiple scans; ILRIS | Point-intercept; forward-looking infrared (FLIR) thermal imaging | Maximum fire temperature and 90th quantile fire temperature; residence time at 300 °C and 500 °C | Longleaf pine (Georgia, USA) |
| Olsoy et al. (2014) [66] | Multiple scans; Riegl VZ-1000 | Point-intercept (non-destructive); harvesting of sagebrush (destructive) | Sagebrush biomass | Sagebrush (Idaho, USA) |
| Calders et al. (2015) [67] | Pre- and post-harvest multiple scans; Riegl VZ-400 | Forest inventory (destructive); tree DBH and height; stem maps; dry weight; AGB | Tree DBH and height; AGB | Eucalypt open forest (Victoria, AU) |
| Rowell et al. (2015) [54] | Multiple scans; Optech ILRIS $3_6$D-HD | Clip plots (destructive); max and mean heights for grass, forbs, shrubs, and litter; mass and weight by fuel type; planar transect counts and fuel bed heights | Fuel bed depths; biomass | Longleaf pine (Florida, USA) |
| Rowell et al. (2016) [57] | Multiple scans; Optech ILRIS $3_6$D-HD | Clip plots (destructive); height; center of mass height; canopy cover; dry biomass by type | Fuel height by type | Longleaf pine (Florida, USA) |
| Cooper et al. [68] | Multiple scans; Compact Biomass LiDAR (CBL) | Disc pasture meter (non-destructive); grass harvesting (destructive) | Grass AGB | Grasslands (South Dakota, USA) |
| Rowell et al. (2020) [53] | Multiple scans; Riegl VZ-2000 | Clip plots (destructive) | Occupied volume and mass; fuel mass; total biomass | Old-field pine-grassland (Georgia, USA) |
| Hillman et al. (2019) [69] | Multiple scans; Trimble TX8 | Field plots with sampling frames (non-destructive) | Vegetation height and cover | Eucalypt (Victoria, AU); Dry sclerophyll eucalypt (Tasmania, AU) |
| Alonso-Rego et al. (2020) [55] | Single scan; FARO Laser Scanner Focus 3D X 130 | 2-by-2 m sampling squares (non-destructive) | Litter depth; shrub cover; mean shrub height; fuel fractions; fuel load | Shrublands (Galicia, Spain) |

**Table 1.** *Cont.*

| Study | TLS Data | Ground Data | Parameters | Landscape |
|---|---|---|---|---|
| Hudak et al. (2020) [58] | Pre- and post-fire multiple scans: LMS 511 horizontal line scanner | Clip plots (destructive); fire consumption of shrubs, grass, and fine downed woody debris; fuel moisture; tree DBH, height, height of crown, and crown diameter | Occupied voxel density; shrub fuel bulk density | Pine (South Carolina, USA) |
| Alonso-Rego et al. (2021) [51] | Single scan; FARO Laser Scanner Focus 3D X 130 | DBH; tree height; base of live crown height; planar transects (non-destructive) | Canopy base height, fuel load, and bulk density; shrub cover; depth of litter and duff; shrub height by species; downed woody debris | Pine (Galicia, Spain) |
| Gallagher et al. (2021) [59] | Pre- and post-fire single scan; Leica BLK360 | CBI by strata; tree height; tree species; DBH (non-destructive) | Substrate, herbaceous, shrub, tree, and total CBI | Pine and pine-oak (New Jersey, USA) |
| Hillman et al. (2021) [70] | Multiple scans; Trimble TX8 | Point-intercept; comparison between multiple sensors | Percent cover; fuel strata classification; canopy fuel height; intermediate canopy height; near-surface fuel height; vertical structure profiles | Dry sclerophyll eucalypt (Tasmania, AU) |
| Pokswinski et al. (2021) [33] | Single scan; Leica BLK360 | Planar-intercept along transects; duff, litter, and fuel bed depths; hourly fuel counts | Reported methodology but did not compare field data and TLS data | Not study-site specific |
| Rodríguez-Lozano et al. (2021) [71] | Multiple scans; Leica ScanStation 2 | Plant height and diameter; green biomass; dry biomass; field spectrometry | AGB; green biomass fraction | Mediterranean steppes (Iberian Peninsula, Europe) |
| Stovall and Atkins (2021) [32] | Multiple scans: Leica BLK360 and Faro Focus 120 3D | None (comparison between two sensors) | Tree DBH, height and total volume; PAVD | Oak-dominant (Virginia, USA) |
| Wallace et al. (2022) [31] | Multiple scans; Trimble TX-8; Faro M70 | Height and percent cover by strata; followed methods of Hines et al. [38] | Height and percent cover by strata | Eucalypt (Victoria, AU) |

AGB = above-ground biomass; DBH = diameters at breast height; CBI = composite burn index; PAVD = plant area volume density.

In summary, traditionally subcanopy, site-level characterization has relied on the time-consuming field methods discussed above. With the availability of cheap (i.e., sub-$30,000 USD) and easy to operate units, practical, operational adoption of TLS for fuel characterization has been suggested and methods have been proposed (see, for example, Stovall and Atkins [32] and Pokswinski et al. [33]). This highlights the need to compare TLS-based assessments with those obtained using traditional field methods.

## 3. Methods

### 3.1. Study Area

The New Jersey PNR is a 400,000 ha landscape comprising predominantly dense, fire-adapted forest types. Much of the PNR is owned by the state government or conservation groups, which has allowed for significant tracts of land to be left undeveloped. The canopy is often dominated by a mix of *Pinus rigida Mill.* (pitch pine) and *Quercus* spp. (oaks), but pure stands of pine or oak are scattered across the landscape as well. Sub-canopy and mid-story species include stunted or immature post oak (*Quercus stellata Wangenh.*), shade suppressed and immature pitch pine, as well as black jack oak (*Quercus marilandica Muenchh*) and mountain laurel (*Kalmia latifolia* L.). Understory species include sheep laurel (*Kalmia angustifolia* L.), shrub oaks such as scrub oak (*Quercus ilicifolia Wangenh.*), and various Ericaceous shrubs (*Vaccinium* spp., *Gaylussacia* spp., and *Lyonia* spp.).

A long history of both wild- and prescribed fires of varying severity has created a mosaic of three-dimensional structure within the forests. Relatively fire bereft areas are characterized by dense woody vegetation, and tracts that experience more frequent wildfire or prescribed fire exhibit unique characteristic patterns of vegetation density due to fire-vegetation feedbacks [72]. Approximately 6000 ha of forest a year are burnt in dormant season prescribed fires, while wildfires burn 3400 ha a year on average. The average frequency and magnitude of wildfires, however, has decreased in the PNR from a peak at the beginning of the 20th century [73,74].

The study area comprised 27 plots of varying fire history scattered among the north–western and north–central portions of the PNR in Burlington and Ocean counties (Figure 1). Plots were chosen to represent either recently burned/prescribed burned conditions or fire excluded conditions. The most frequently burned plots experienced prescribed fire on an annual basis, while several fire suppressed plots have been without fire for more than 25 years, with the longest time since a burn being 91 years. Canopy tree species among the plots were predominantly pitch pine, with a single plot dominated by chestnut oak (*Quercus montana Willd*.). Mid-canopy and understory species included those mentioned previously in this section.

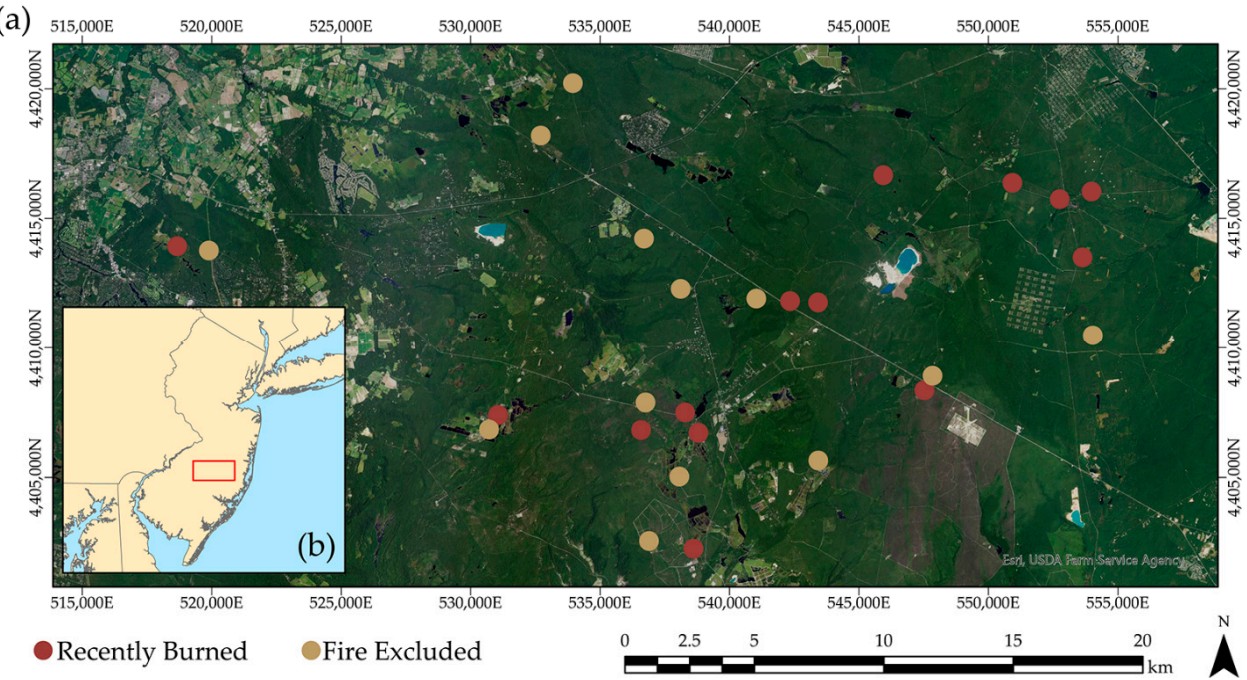

**Figure 1.** Study area in New Jersey Pinelands National Reserve (PNR). (**a**) Location of field plots. (**b**) Location of (**a**) within New Jersey, USA. Coordinates in (**a**) are relative to the NAD83 UTM Zone 17N projection. Both maps are projected to the NAD83 UTM Zone 17N projection. Base imagery is from the National Agriculture Imagery Program (NAIP) of the United States Department of Agriculture (USDA) Farm Service Agency.

### 3.2. TLS and Ground Data

Each of the 27 plots was circular in shape and had a radius of 10 m. This study specifically used 10 m radius plots since the goal was to characterize local, plot-level characteristics and because of the limitations of single-position, single-scan TLS data. Due to issues of occlusion, decreasing point density with radial distance from the scanner, and beam divergence, Pokswinski et al. [33] suggested that only data within 10 to 15 m of the Leica BLK 360 TLS model used in this study are usable. To collect field reference data, each plot was sampled 40 times within the area defined by the 10 m radius from the plot center and measured using a Lufkin foldable 2 m wood rule. The 40 measurement locations within each plot were chosen randomly within the 10 m radius by tossing metal flagging and measuring the height of the tallest woody vegetation, in centimeters, where the flagging landed. If flagging landed in an area with no woody vegetation, that sample was recorded as 0 cm. The goal was to collect a total of 40 randomly selected reference data points within each of the 27 plots in order to generate an unbiased sample of shrub heights within the 10 m radius being characterized.

TLS data were collected using a Leica BLK 360 using a similar approach as Gallagher et al. [59]. Two single-return scans were conducted at each plot, one scan during the dormant season and another during the growing season, in order to record scans in both leaf-off and leaf-on conditions. Scans were taken with the TLS positioned at the plot center. Unprocessed scans were downloaded using the Leica BLK Data Manager and extracted as PTX files using the Leica Cyclone Register 360 software.

### 3.3. Data Preparation

Figure 2 conceptualizes the data preparation, modeling, and assessment and comparison methods used in this study. All processing, modeling, and assessment were conducted within the R [75] data science language and environment. We specifically made use of the lidR [76] and TreeLS [77] packages, which allow for reading and processing LiDAR and TLS

data in the R environment. Since all shrub height field measurements were collected within a 10 m radius of the sensor, returns occurring within this radius were extracted from the larger dataset. Next, a noise filter was applied based on Z value distributions. Following the noise filtering, ground classification was performed using the cloth simulation function (CSF) as implemented in the lidR [76] and RCSF [78] packages. In order to classify ground returns, the CSF models a rigid cloth surface, which is defined by interconnected nodes within a three-dimensional space. In other words, the point cloud is inverted, and the cloth surface is modeled above the points. The nodes associated with this modeled surface are then used to classify ground return points while honoring rigidness constraints. We used a class threshold of 0.05 and a cloth resolution of 0.05; the class threshold specifies the distance threshold to the simulated cloth to classify a point as a ground return, while the cloth resolution relates to the distance between nodes making up the modeled cloth surface. Rigidness was maintained as the default value of 1 to allow for the modeling of rugged terrain, and the time step, which relates to how gravity is simulated in the model, was set to 0.65.

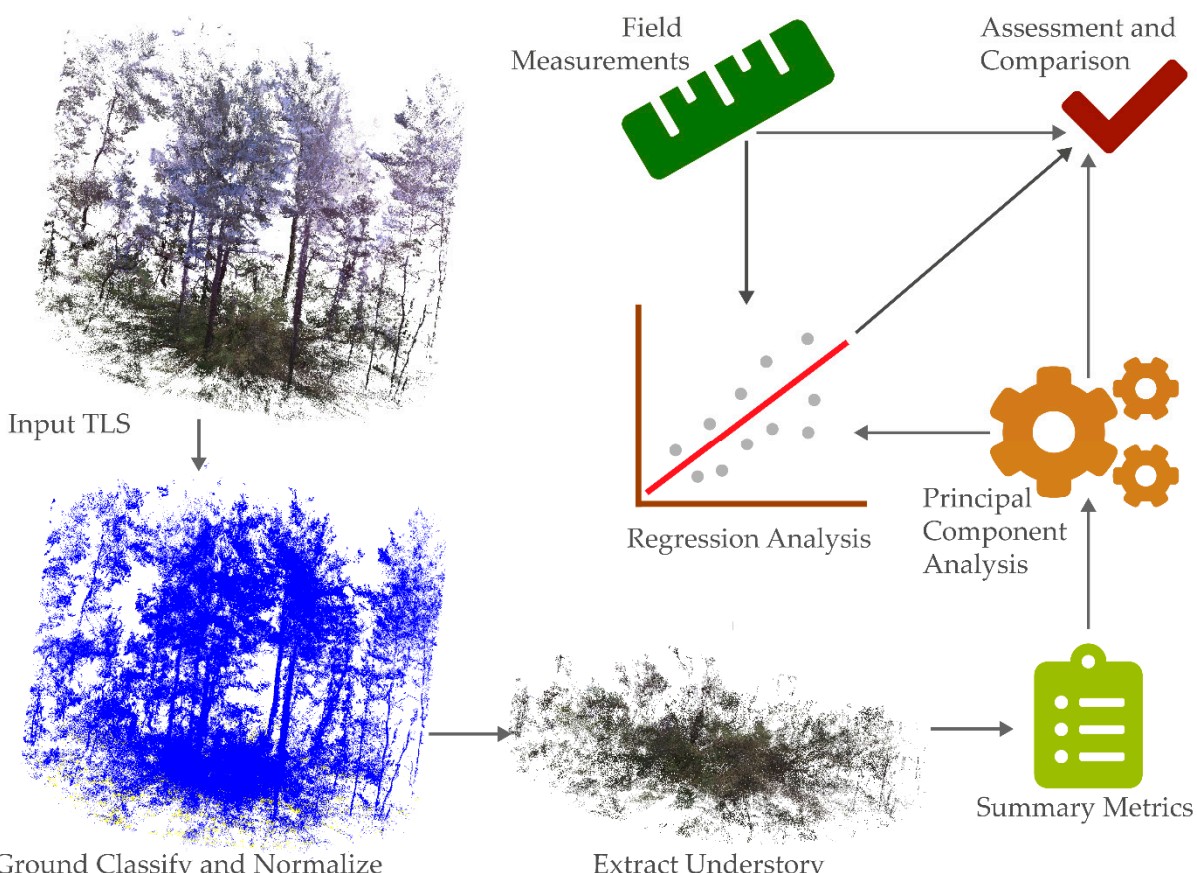

**Figure 2.** Conceptualization of data preparation, modeling, and assessment and comparison workflows. Icons from Font Awesome and made available under a CC by 4.0 license.

Next, lidR [76] and TreeLS [77] were used to normalize the point cloud to convert the Z coordinates to height above ground, as represented by the ground point classification created using the CSF method, and minimize the impact of variable topography. This first required that a triangulated irregular network (TIN) be created from the classified ground returns, followed by a rasterization of this surface to create a digital terrain model (DTM) of bare earth surface elevations at a 0.1 m spatial resolution. The ground elevation measurements from the DTM surface were then subtracted from the returns occurring above them. Since we were specifically interested in estimating shrub heights, returns associated with trees were detected then removed from the point cloud using the TreeLS [77]

package and the methods described by de Conto et al. [77]. This method specifically relies on a cylinder fitting technique to identify main stems or trunks. Once main stems are detected, points are assigned to individual trees using a graph theory approach. All points that are associated with a specific tree are assigned the same unique tree identifier, allowing for selecting each individual tree, or, as was conducted in this study, filtering out all points returning from trees and associated stems or leaves. This ground classification followed by tree segmentation workflow resulted in a ground normalized point cloud containing only points classified as understory returns.

Once the TLS data were filtered for noise, clipped, and ground classified and trees were classified and removed, it was assumed that the remaining points represented returns from the understory. These understory returns were used to extract a set of 54 metrics to summarize the understory conditions within the plot extent. The metrics generated are listed in Table 2. For all the non-ground/understory returns, we calculated the mean, median, standard deviation, skewness, and kurtosis of the $Z$, or height, values. We also calculated the $Z$ values associated with all deciles between 1 and 9. The point cloud was also segmented into height strata as follows: >0 to 0.5 m, 0.5 to 1 m, 1 to 1.5 m, 1.5 to 2 m, and >2 m. This was accomplished using the normalized $Z$ values, and these bin ranges were selected since our primary interest was characterizing subcanopy features smaller than 2 m in height. Within each height strata, we counted the number of returns and the percentage of all non-ground/understory returns returning in the height strata. We calculated the mean, median, standard deviation, skewness, and kurtosis of the $Z$ values within each height bin. Lastly, and in order to summarize the distribution of points relative to the $X/Y$ plane, we calculated the average nearest neighbor (ANN) index, which offers a measure of the spatial dispersion or clustering of a point pattern, using the spatialEco package [79].

**Table 2.** Metrics calculated from TLS point cloud data following ground classification, height normalization, and extraction of non-ground/understory returns.

| Group | Variables | Count |
|---|---|---|
| All non-ground/understory returns ($Z$) | Z mean, median, standard deviation, skewness, and kurtosis | 5 |
| | Deciles (1–9) | 9 |
| Strata-based | Count of returns in strata, percent of all non-ground/understory returns in strata | 10 |
| Strata-based ($Z$) | Mean, median, standard deviation, skewness, and kurtosis | 25 |
| Strata-based ($X/Y$) | Average nearest neighbor (ANN) index | 5 |
| Total | | 54 |

*3.4. Regression Modeling*

Models were generated using multiple linear regression fitted using the ordinary least squares (OLS) method [80] as implemented in the stats base R package [75]. Due to the limited number of samples, it was not possible to partition the data into separate training and testing sets. Instead, models were trained using all available samples but withholding one sample. The withheld sample was then predicted using the model trained using the other samples. This process was repeated such that all samples were held out and predicted using a model trained with all other available samples. Additionally, due to the limited sample size relative to the number of predictor variables, the predictor variables were transformed into eight new and decorrelated predictors using principal component analysis (PCA) [81]. The first eight principal components were chosen as it was found that this was adequate to capture 99% of the variance in the original 54 variables. So as not to cause a data leak, the PCA was performed separately for each model with one of the samples held out. As a result, the first eight principal component variables were not consistent between models since they were generated using different training and

validation partitions. Other than the eight principal component variables, we also included whether the site was recently burned or fire excluded, represented as a single dummy variable, as an additional variable in the analysis, resulting in a total of 9 predictor variables in each model.

In order to assess the impact of reducing the number of shrub height samples collected within each plot, we randomly selected 2 through 40 with a step size of 2 random samples, calculated the mean shrub heights in each plot using the subsample, then used this estimate as the dependent variable in the multiple regression analysis. For the withheld sample in each model run, the total set of 40 samples was always used to estimate the average shrub height in the plot. Separate models were also developed using the leaf-off and leaf-on data. Generating models using sample sizes varying from 2 to 40 with a step size of 2, trained on all but one sample, and with the leaf-on and leaf-off predictor variables resulted in a total of 1080 models.

### 3.5. Assessment and Comparison

Modeling results were assessed and compared using the prediction R-squared and root mean square error (RMSE) metrics [80], which were calculated using the shrub heights estimated using all 40 samples available in each plot and the associated prediction for each sample when it was held out of the modeling process. RMSE was calculated in the units of the response variable, in this case cm. In order to assess and compare changes in the precision or uncertainty of the predictions, we also calculated prediction intervals (PIs) using a 95% confidence interval [82]. Note that precision intervals are different from confidence intervals, as they are calculated for each prediction as opposed to the entire model.

## 4. Results and Discussion

### 4.1. Distribution of Shrub Heights within Plots

Figure 3 summarizes the central tendency and distribution of the 40 field-based shrub height measurements collected within each of the 27 plots. Generally, sites that had been recently burned showed lower average shrub heights in comparison to fire excluded sites. Measured shrub heights were rarely above 2 m within recently burned sites. Additionally, shrub heights were generally more variable within fire excluded than in recently burned plots. Both a one-tailed *T*-Test with unequal variance (*p*-value = 0.0001) and a one-tailed Wilcoxon rank sum exact test (*p*-value = 0.0002) suggested that the recently burned sites had a statistically significant lower mean shrub height at the 95% confidence level in comparison with the fire excluded sites. Further, a one-tailed *T*-Test with unequal variance (*p*-value = 0.0001) and a one-tailed Wilcoxon rank sum exact test (*p*-value = 0.001) both suggest that the recently burned sites had statistically significantly lower variability of measured heights within the plots in comparison with the fire excluded sites. This highlights the value of including the recently burned vs. fire excluded variable in the multiple regression models.

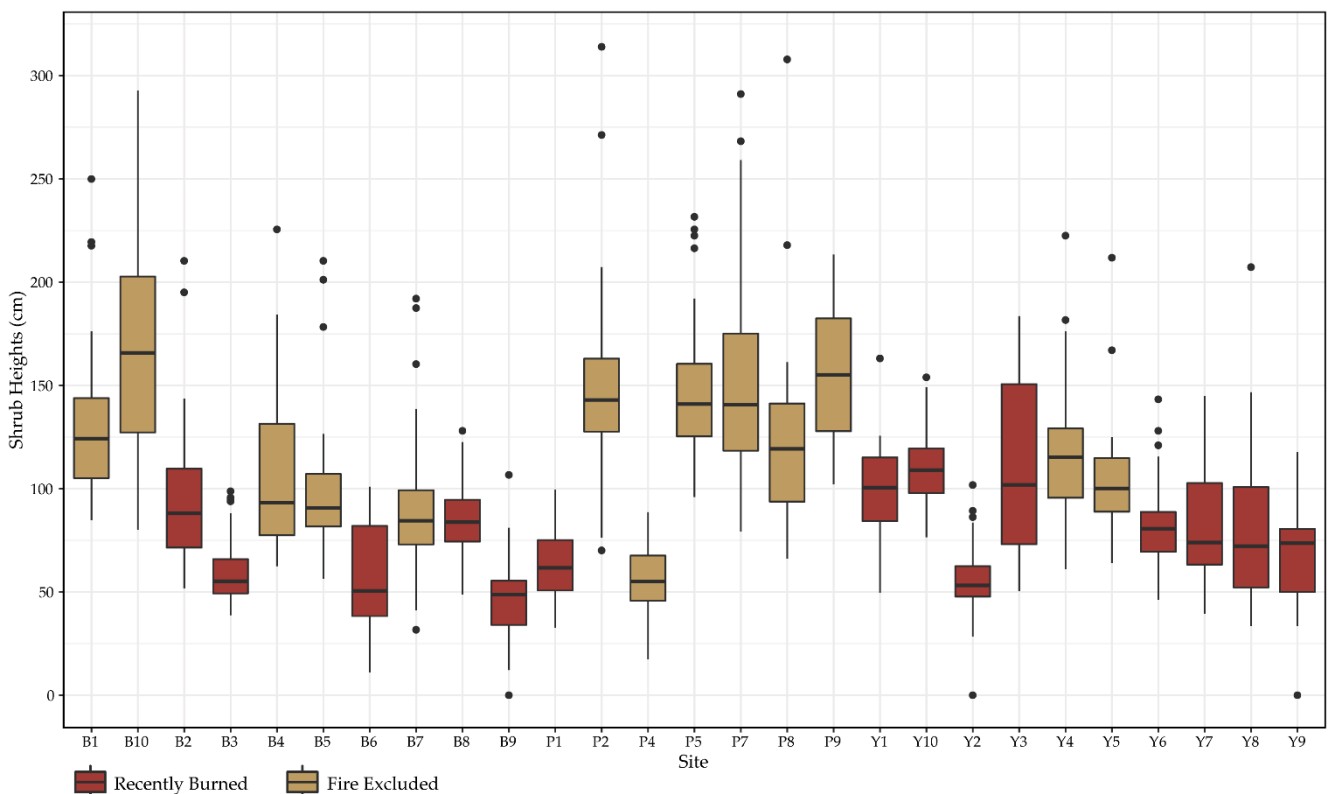

**Figure 3.** Distribution of 40 field-measured shrub heights in each of the 27 plots. Color differentiates recently burned and fire excluded sites.

*4.2. Impact of Number of Shrub Height Measurements and Phenology on Model Performance*

Figure 4 shows the predicted shrub heights for each plot when using all 40 field samples to estimate the mean shrub height. As described above, each site was predicted using a model trained with all other available plots. Black points represent the mean shrub height calculated from the 40 field samples, while the colored points represent the predicted mean shrub height obtained using the multiple linear regression model. The error bars represent the 95% confidence PIs. The mean PI across all of the sites was 72.9 cm when using the leaf-off metrics and 54.6 cm when using the leaf-on metrics. When using the leaf-off metrics, the average prediction interval for recently burned sites was 79.4 cm, while it was 65.8 for fire excluded sites. When using the leaf-on metrics, the average prediction interval for recently burned sites was 57.5 cm, while it was 51.4 for fire excluded sites. The predicted R-squared calculated using the residual for each sample withheld from each model was 0.78 when using the leaf-off metrics and 0.89 when using the leaf-on metrics. The RMSE was 12.9 cm and 10.3 cm for the leaf-off and leaf-on models, respectively.

Generally, the results suggest better performance when using the metrics derived using the leaf-on data in comparison with the metrics derived using the leaf-off data. The reasons for the differences in performance with phenology are not clear; however, this may relate to having a greater density of understory returns, and, as a result, a better characterization of the understory, in the leaf-on data due to the presence of foliage. However, there are some confounding variables. Specifically, it would be expected to have a less accurate ground classification and subsequent ground normalization in the leaf-on vs. leaf-off data due to a lower density of ground returns due to potentially more ground occlusion by vegetation. The models also generally provided lower prediction intervals for fire excluded vs. recently burned sites. This may partially be related to differences in mean heights and variability in heights within plots between the recently burned and fire excluded groups.

These results specifically highlight the value of providing PIs along with each prediction. For example, users can use these values to determine whether predictions are precise

enough to meet management, reporting, or modeling standards. Additionally, PIs allow for better quantification and understanding of differences in prediction variability between sites or groups of sites (i.e., recently burned vs. fire excluded).

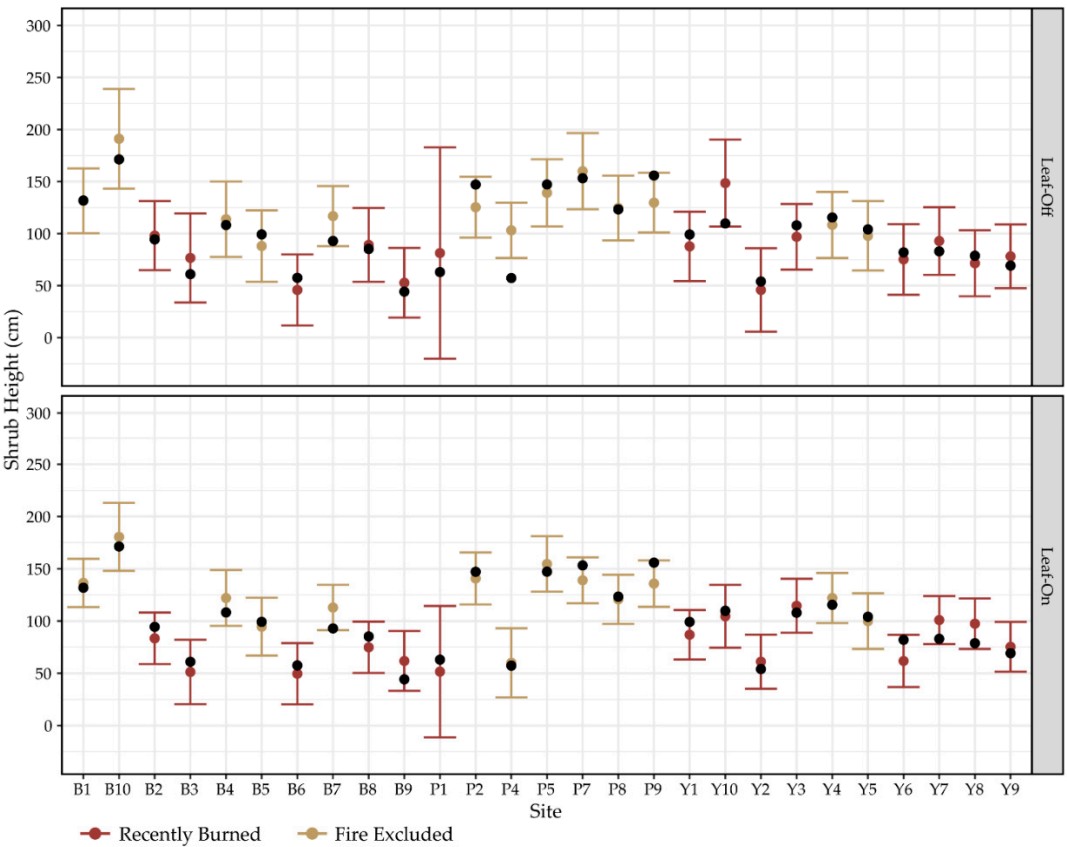

**Figure 4.** Mean shrub heights based on 40 ground samples (black dots) and predicted shrub heights obtained using multiple linear regression and TLS metrics calculated from leaf-off and leaf-on data. Error bars provide the 95% confidence PI for each plot.

Figure 5 summarizes variability in model performance as the number of samples within each plot used to estimate the mean shrub height in the plot are varied from 2 to 40 m with a step size of 2 m. Figure 5a shows the mean PI by sample size, Figure 5b shows the mean RMSE, and Figure 5c shows the mean prediction R-squared. All three metrics generally suggest stronger performance when using the leaf-on metrics as opposed to the leaf-off metrics, as noted above for the results obtained when using all available field samples. These results generally suggest that model performance is sensitive to the number of field reference samples available in each plot, regardless of whether leaf-on or leaf-off data are used. Results were notably poorer when less than 10 samples were collected within each plot. Performance generally stabilized after 20 samples were used to calculate plot means. We attribute fluctuations in the model performance with changes in sample size to be partially related to the random samples chosen to train the models. More specifically, more variability in model performance would be anticipated with a smaller sample size, since which samples were chosen to train the model would impact the resulting prediction. With an increase in the number of training samples, it is anticipated that a more consistent result would be obtained between model runs. Similar to the results reported by Keane et al. [38] when comparing the planar intercept, fixed-area microplot, and photoload field methods, our results suggest that intensive sampling is necessary to accurately estimate fuel conditions. In this study, collecting only a few samples per plot (i.e., less than 10) was inadequate to provide stable and precise estimates of mean shrub heights within 10 m radius plots when used as a dependent variable to train multiple

regression models to estimate this fuel metric from TLS-derived metric sets, both leaf-off and leaf-on, as measured using mean PIs, RMSE, and R-squared. In other words, field reference sampling density is an important consideration when these data will be used to train models to make estimates from remotely sensed data for application to new data within other plots or extents.

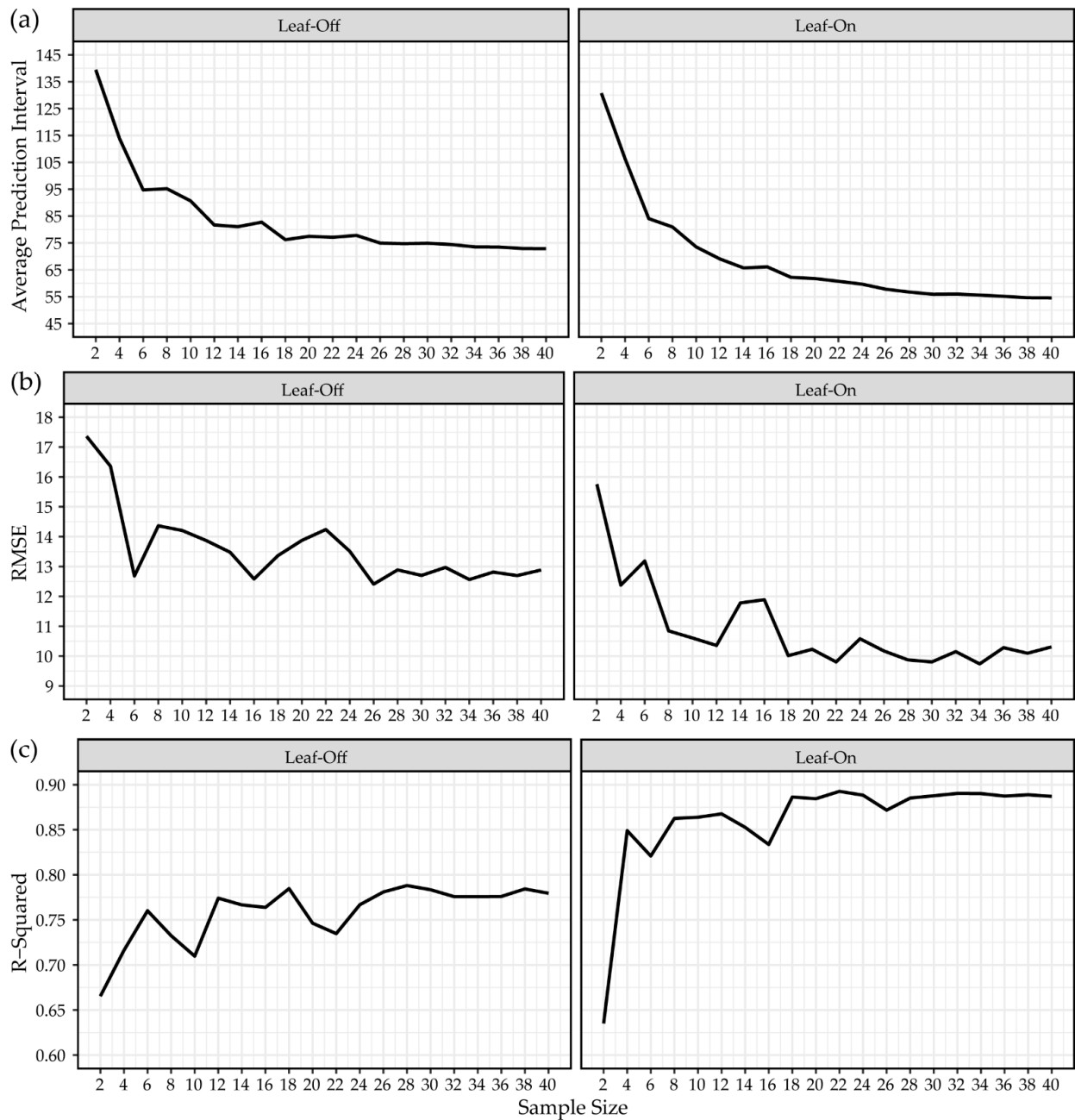

**Figure 5.** Changes in model performance with field reference data sample size. (**a**) Mean PI; (**b**) RMSE; (**c**) prediction R-squared.

There are some notable limitations in this study. First, although we had 40 random field measurements available within each plot, only 27 plots were available. In order to minimize the impact of a small sample size, we generated multiple models and withheld each sample for validation so as not to require partitioning the data into separate training and testing sets. We also relied on multiple linear regression, as opposed to a machine learning modeling method that may be more likely to overfit to a small dataset. The findings of this study

may not be applicable to all forested landscapes. This study was conducted within the New Jersey PNR, and results may not extrapolate to other forest types or landscapes with disparate community composition, understory and/or overstory characteristics, and/or burn histories. For example, less field samples may be necessary within forest stands with more homogeneous understory characteristics or less variable shrub heights. In contrast, forests with more heterogeneous understory characteristics may require a larger sample size. The types of species making up the understory and the disturbance or management histories may also have an impact. Further investigation of the impact of reference data sampling density on plot-level model performance in other landscapes would be useful. Lastly, we relied on single-return, single-scan TLS data collected with cheaper units since our focus was on operational adoption for rapid plot-level assessment. Results may vary when using other sensors or combining multiple scans to more fully characterize a plot or minimize occlusion. Even given these limitations, we argue that our results highlight the importance of field reference sampling density and data phenology when such data will be used as input to empirical predictive modeling routines.

To improve the mapping and characterization of fuels, especially in the forest understory, there is a need for further development of methods for segmenting point clouds into different types of fuels and mapping or differentiating individual shrubs or trees [83]. Quantitative structure models (QSMs) could be especially useful for such segmentation work [84–86]. Many factors can impact fire behavior, including abundance and composition of duff and detritus, abundance and distribution of live and dead woody material, structural elements and configuration of the canopy and subcanopy, and foliage abundance and characteristics [1,2]. Adopting methods from other disciplines or areas of research may be of particular value for improving the characterization of fuels; for example, computer graphics and porous media theory methods have been proposed to better model the gap fraction of tree crowns [87]. As noted by White et al. [83], there is a need to develop best practices for using TLS data to estimate plot-level attributes. Using these data to enhance forest inventories remains challenging due to lingering technological, methodological, and operational issues. For example, methods to accurately and consistently measure key forest inventory attributes, such as number of trees, species, diameter at breast height, and height, are still lacking [83].

## 5. Conclusions

This study highlights the importance of field reference data sampling density when they are used to derive a dependent variable for empirical modeling of plot-level characteristics using remotely sensed data. Specifically, we document reductions in mean PIs, RMSE, and R-squared with reduction in the number of reference samples available within each plot for estimating mean plot-level shrub heights using metrics derived from TLS data. Models were generally less accurate and precise when less than 10 random shrub heights were collected within each plot, but stabilized after 20 samples were available. Additionally, metrics derived from leaf-on TLS data generally provided more accurate and precise predictions than those calculated from leaf-off data within the study plots and landscape. This study highlights the importance of field reference sampling design and data characteristics when data will be used to train empirical models for extrapolation to new sites or plots.

**Author Contributions:** Conceptualization, A.E.M., M.R.G., E.L.L. and N.S.S.; data curation, M.R.G., N.M. and S.M.P.; formal analysis, A.E.M., M.R.G., M.S.B. and S.M.P.; funding acquisition, M.R.G. and N.S.S.; investigation, A.E.M., M.R.G., N.M. and M.S.B.; supervision, M.R.G. and N.S.S.; writing—original draft preparation, A.E.M. and M.R.G.; validation, A.E.M. and M.R.G.; writing—review and editing, A.E.M., M.R.G., N.M., M.S.B., S.M.P., E.L.L. and N.S.S. All authors have read and agreed to the published version of the manuscript.

**Funding:** Funding was provided by the National Science Foundation (NSF) (Award Number: 2040676, "NSF Convergence Accelerator Track D: Artificial Intelligence and Community Driven Wildland Fire

Innovation via a WIFIRE Commons Infrastructure for Data and Model Sharing"). Any opinions, findings, and conclusions or recommendations expressed in this material are those of the author(s) and do not necessarily reflect the views of the National Science Foundation. This project was also funded by the USDA Forest Service Northern Research Station through joint venture agreement 20-JV-11242306-069 and the US Fish and Wildlife Service under grant number F21AC02192-00.

**Data Availability Statement:** Data and code associated with this study are made available on the WV View webpage (https://www.wvview.org/research.html, accessed on 17 February 2023).

**Acknowledgments:** We would like to thank the three anonymous reviewers whose comments strengthened the work.

**Conflicts of Interest:** The authors declare no conflict of interest.

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
