# Peer review of "Impact of Reference Data Sampling Density for Estimating Plot-Level Average Shrub Heights Using Terrestrial Laser Scanning Data"

_fire, doi:10.3390/fire6030098_

Round 1
Reviewer 1 Report
In this paper, the Terrestrial Laser Scanning data and field survey data are used to discuss the impact of forest phenology and density of shrub height measurements on estimating average shrub heights at the plot level using multiple linear, regression and metrics derived from ground-classified and normalized point clouds. The final results showed that shrub sampling density had no significant effect on the estimation of irrigation in the study plot. The whole framework of the paper is meaningful.
There are some scientific problems that deserve further consideration by the authors:
1) Section 3.3, line 309, in this part of the study, how to conduct the height division for point clouds to obtain shrub components from the ground LiDAR data set should be elaborated.
2) In Section 3.4, line 329, please afford the specific eight principal component variables.
3) Section 4.2, line 445, figure 5 shows that in the leaf opening and leaf closing measures of (b) and (c), RMSE and R square both have relatively large fluctuations several times. What might be the cause of such changes? It should be detailed and illustrated.
4) In the conclusion section, the limitations of this method, optimal direction and future research prospects of the method can also be discussed.
5) Some factors, e.g., deciduous leaves deposited on the ground, woody components for each tree species, vertical structures and foliage element distribution of the forest canopies, will affect fuel characteristics and fire possibility. A more fine work of characterizing tree and forest phenotypic traits named “A reinterpretation of the gap fraction of tree crowns from the perspectives of computer graphics and porous media theory” might benefit the future research direction, which should be mentioned in the Discussion section.
Reviewer 2 Report
Please see the attached

Reviewer 3 Report
Dear Authors,
I have reviewed the paper titled: “Impact of Reference Data Sampling Density for Estimating Plot-Level Average Shrub Heights using Terrestrial Laser Scanning Data". In my opinion, the aims of the paper are germane with “Fire” journal topic, in the present form, the paper fits with the international scientific standards. The paper is written with an acceptable English level. The contribution of this paper to the scientific knowledge is good, only few minor issues are present. I understand the difficult work done, but as a reviewer it is my duty to highlight the gaps in order to improve the research approach and its presentation to the international scientific community. Please I suggest revising the paper following the suggestions reported in the file attached.

Round 2
Reviewer 1 Report
As a whole, I feel that the revisions and additions to the manuscript add value to both its readability and scientific merit. The authors did a nice job of estimation plot-level average shrub heights using terrestrial laser scanning data. The manuscript is well written, and the revisions made to the current draft make it better suited for publication.
Reviewer 2 Report
I feel the revised version is OK to be published.